# Ladders of Authority, Status, Responsibility and Ideology: Toward a Typology of Hierarchy in Social Systems

**A. Georges L. Romme** 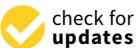

Department of Industrial Engineering & Innovation Sciences, Eindhoven University of Technology, P.O. Box 513, 5600 MB Eindhoven, The Netherlands; a.g.l.romme@tue.nl; Tel.: +31-40-247-2170

**Abstract:** Hierarchy is a key characteristic of any complex system. This paper explores which notions of hierarchy are being used in the field of organization and management studies. Four distinct types of hierarchy are identified: a ladder of formal decision-making authority, a ladder of achieved status, a self-organized ladder of responsibility and an ideology-based ladder. A social mechanism-based perspective serves to define and distinguish these four types. Subsequently, the typology is further developed by comparing the four hierarchy types in terms of their tacit/explicitness, (in)transitivity and behavior- versus cognition-centeredness. This article contributes to the literature by dissecting the general metaphor of hierarchy into four different constructs and their social mechanisms, which serves to create a typology of the various ways in which complex social systems can be characterized as hierarchical. This typology can inform future research drawing on any type of hierarchy.

**Keywords:** hierarchy; complexity; organization; social system; formal authority; social mechanism; self-organization; responsibility; status; typology

## 1. Introduction

The notion of hierarchy is widely used but is also rather ambiguous because highly different interpretations exist. For example, the notion of hierarchy in software systems refers to different levels of abstraction—such as those in an (e.g., Android) operating system [1]. In companies and other organizations, hierarchy often involves a sequence of levels of formal decision-making authority [2–4]. Another notion is hierarchy as a ladder of ideology, in which people establish themselves as legitimate leaders by invoking some (e.g., religious or political) idea to legitimize the relationship between higher or lower levels [5]. Yet another notion has been developed in the literature on organization design and organizational agility, which conceives of hierarchy as a requisite system that emerges in a self-organized manner from operational activities [6–9].

Herbert Simon acknowledged the generic nature of the notion of hierarchy, arguing that all complex (e.g., natural or social) systems "consist of a hierarchy of components, such that, at any level of the hierarchy, the rates of interaction within components at that level are much higher than the rates of interaction between different components" [10] (p. 587). Despite the broad applicability of Simon's perspective on hierarchy, the hierarchical nature of social systems has only been theorized in terms of the distinction between formal and informal hierarchy [11,12]. However, this formal-informal dichotomy does not cover the entire landscape of how hierarchy has been conceptualized and instantiated (e.g., [5–7]). Therefore, the purpose of this article is to develop a typology of hierarchy, by mapping the various ways in which hierarchy is defined.

Based on an extensive review of the literature, four types of hierarchy are identified: a ladder of formal decision-making authority levels, also known as formal hierarchy [2,3]; a ladder of achieved status levels (e.g., arising from seniority or expertise), also known as informal hierarchy [13,14]; a ladder of responsibility levels, arising from self-organizing initiatives throughout an organization or another social system [6,7]; and a ladder of

ideology drawing on a set of shared beliefs to justify the relationships between higher and lower levels [5]. Subsequently, this typology is further developed by comparing the four types on several key dimensions.

This paper contributes to the literature by dissecting the general metaphor of hierarchy into four fundamentally different constructs. The resulting typology clarifies the central role of the hierarchy construct for any complex social system, by defining the distinct mechanisms of decision-making authority, achieved status, self-organized responsibility and strong ideology.

## 2. Review Scope and Approach

Earlier reviews [11,12] in this area were instrumental in defining the notions of formal and informal hierarchy. These reviews and related studies (e.g., [15]) suggest that formal and informal hierarchies tend to complement each other. As argued in the first section, these prior reviews have not mapped the entire landscape of hierarchy as a key characteristic of complex systems, and therefore a more inclusive taxonomy and typology is developed in this article.

The review in this paper largely focuses on organizational systems because an initial literature search demonstrated that the hierarchy notion is most frequently used in the field of organization and management studies. The subsequent search for relevant publications was conducted in September 2020, using queries like "hierarchy" in combination with "manag*" and/or "organ*" in Google Scholar (https://scholar.google.com) and Web of Science (https://clarivate.com/webofsciencegroup/solutions/web-of-science). After filtering out studies in other domains (like information systems) as well as publications referring to hierarchy in the context of research methods or other tools, sources were further filtered using the criterion that any publication selected would need to address hierarchy as a core construct. The combined use of Web of Science and Google Scholar was instrumental in locating articles in double-blind-reviewed journals as well as widely cited monographs and books (e.g., [16,17]) in the field of organization and management studies. I only added sources from adjacent disciplines like sociology and law when the results of (reviewing) the initial set of sources pointed at the need to consult these additional publications. This also implies that the next section contains several examples of (hierarchy used in) social systems other than organizations.

A complete overview of the literature would entail a database of more than 10,000 publications. Therefore, I adopted an iterative approach in which publications were continually added (to an excel file containing outlines of each publication) and reviewed until a saturation effect was observed; that is, I stopped adding publications when the last 25 publications added to the database did not point at any new types and/or social mechanisms of hierarchy. Each publication was reviewed and coded regarding the definition, social mechanisms and assumptions of hierarchy. This results in a total of 190 publications underpinning the typology described in the next section, of which 75 sources are referenced in this article.

One implication of the review approach adopted is that hierarchy notions at the micro-level (e.g., individual and group behavior), as well as the macro-level (e.g., strategy, organization design), are included, while also exploring various related bodies of literature—for example, the literature on requisite structure [6]. Moreover, in defining and comparing the various types of hierarchy (for a preview, see Table 1), I adopt a *mechanism-based* perspective [18,19]. In this respect, social mechanisms such as the "social construction of status" are instrumental in explaining why a specific hierarchy type arises and/or prevails [20]. The notion of social mechanism has been previously used to bridge and synthesize insights from different philosophical perspectives and research streams (e.g., [21,22]) because it is relatively agnostic about the nature of social action and can, therefore, steer a path between positivist, narrative and functional perspectives [18]. This agnostic lens is important here, because the notion of hierarchy is used in fundamentally different paradigms and discourses (e.g., [2,23–25]).

### 3. Main Findings: Four Types of Hierarchy

*3.1. Hierarchy as Ladder of Authority*

A common conception of hierarchy is to define it as a sequence of levels of formal authority, that is, the authority to make decisions [3,4,26–29]. Following Weber [30], a ladder of authority involves the vertical formal integration of official positions within a single organizational structure, in which each position is under the supervision and control of a higher one. Similarly, Dumont refers to "a ladder of *command* in which the lower rungs are encompassed in the higher ones in regular succession" [17] (p. 65). This results in a ladder that systematically differentiates authority, for example: the board of directors, CEO, division managers, heads of department, team leaders and operational workers.

The social mechanism driving any ladder of authority is the legitimacy of decision-making authority, which arises from the constitution or statutes (or any other legal structure) of the incumbent organization—regardless of whether this is a company, non-profit organization or governmental agency [31]. A key constitutional or statutory principle is that people at the top of the authority ladder, as rightful holders of the power to make key decisions, have the right to dictate targets and/or behaviors and are entitled to be obeyed [31]. Accordingly, decision-making authority is, at least initially, concentrated at the top of the ladder [7]. The top echelon can then delegate specific decision authorities to lower levels—also in view of the limits on the amount of attention (cf., bounded rationality) that top managers can give to various issues [3,4]. In this respect, organizational systems like companies, associations, clubs and municipalities tend to have elaborate constitutions that contain the fundamental principles and bylaws regarding positions, decision domains and related issues [31].

In many publicly traded companies, ownership and control have become largely separated [31,32]. As a result, the ladder of authority in these companies has become rather complex, in terms of the formal authority arising from the shareholders' legal ownership, the CEO controlling the company on a day-to-day basis, and non-executive directors engaging in supervisory activities [33–36].

*3.2. Hierarchy as Ladder of Status*

Another widely used meaning of hierarchy is in terms of informal or unofficial mechanisms to rank people [14,23]. These informal mechanisms are highly person-dependent, involving, for example, social norms and values, verbal or non-verbal attitudes and behaviors and guidelines for communication [11]. At a more fundamental level, the source of these informal hierarchies is differences in personal status, other than those arising from formal authority. Status is one's social standing or professional position, relative to those of others [37] or "the respect one has in the eyes of others" [12] (p. 351). In anthropology and sociology, this notion of status is also known as "achieved status", the social position that is earned, instead of being ascribed [38,39]. The underlying mechanism is social stratification, a social mechanism that draws on shared cultural beliefs that can make status differences between people appear natural and fair [40,41].

Ladders of status are frequently observed in empirical work (e.g., [13,14,23]). For instance, He and Huang [23] studied how the deference for one another gives rise to a status hierarchy within a firm's board of directors. Another example is Dwertmann and Boehm's study [13] of how status drives the quality of the relationship between supervisor and subordinate. Overall, any ladder of status is socially constructed, which makes it fundamentally different from the ladder of authority that (largely) arises from the legal structure of the organization. This also implies that a status ladder is much more fluid and adaptive than its authority-driven counterpart.

While social comparison can to some extent also take place between (people from) different units and departments within an organization [42], the person-dependent nature of status implies that ladders of status primarily arise within the group of people one interacts with on a daily basis—be it a team, work unit, department or network of people [13,14,23,37,43].

### 3.3. Hierarchy as Ladder of Responsibility

In the literature on organization design and organizational agility, hierarchy is conceived as a requisite structure that emerges in a self-organized manner from operational activities [6–8]. For example, Jaques [6,44] argued hierarchy is the only effective organizational mechanism that can employ large numbers of people and yet preserve unambiguous accountability for the work they do. Jaques' notion of hierarchy is part of his broader perspective on requisite organization, defined as the organizational roles and connections that make the entire system operate efficiently as required by the nature of human nature and the enhancement of mutual trust [6]. The notion of requisite hierarchy has informed the development of new organizational forms like holacracy [8,45], which involves a system of self-organizing circles that structure roles and work processes [7,9]. In designing holacracy, Robertson [45] assumed that this hierarchical network of circles, at any given point in time, has an (ideal) requisite structure that "wants" to emerge.

More specifically, the key mechanism driving hierarchy in these agile and/or holacratic forms of organizing is that agents at all levels self-organize their responsibility, that is, exercise "real" rather than formal authority [7,46]. In this respect, responsibility is an expression of self-restraint and intrinsic obligation [47–49]. Other examples of self-organized ladders of responsibility have been observed in (the early stages of) worker cooperatives in which hierarchy is created in a bottom-up manner [50] and in so-called sociocratic organizations that draw on a circular hierarchy of double-linked circles [51].

### 3.4. Hierarchy as Ladder of Ideology

The fourth conception of hierarchy identified in the literature is the so-called ladder of ideology, in which people establish themselves as legitimate leaders by invoking some (e.g., religious, spiritual or political) idea to legitimize the hierarchical relationship between higher or lower levels [5,52–54]. Ideological hierarchies have a long history, for example in the form of the administrative hierarchies headed by pharaohs in ancient Egypt or those headed by kings in medieval Europe. The main legitimacy of any pharaoh or king arose from the strong belief in the idea that the pharaoh/king acts as the intermediary between the gods and the people, and thus deputizes for the gods [54].

Another example is the hierarchy prevailing until today in the Balinese community, which is strongly connected to the rice cycle that is believed to constitute a hierarchical relationship between gods and humans, both of whom must play their parts to secure a good crop; the same ideology also legitimizes the hierarchical relationship between high castes and low castes in Bali [53]. Similarly, ideological ladders have long been sustaining the way the Catholic church, as well as the Hindu caste system, operates. For example in the case of Hinduism, "the religious justification for the four core castes lies in the belief that each derived from a different part of the creator God's (Brahma) body, descending from the head downwards" [55] (p. 31).

Ladders of ideology continue to exist in many settings; for instance, ladders fueled by prevailing values and beliefs among members of the organization about how the world should operate [56,57]. For example, Brummans et al. [5] identified a ladder of ideology in their study of how leaders in a Buddhist humanitarian organization create and sustain hierarchical relationships with subordinates. They observed that these leaders invoke a spiritual entity in their daily interactions and use this invocation to direct their organization and establish a shared sense of compassion and wisdom [5]. Another example is the ideology of "shareholder value maximization" that is widely used in publicly owned corporations [58]; this ideology helps to create and sustain the image of a clear hierarchy from shareholders to employees—although, in practice, the separation of legal ownership and actual control implies that the CEO together with the Board of Directors are, in fact, at the top of the corporate hierarchy [31,32].

**Table 1.** An overview of four types of hierarchy.

|  | **Ladder of Authority** | **Ladder of Status** | **Ladder of Responsibility** | **Ladder of Ideology** |
|---|---|---|---|---|
| *Definition* | Sequence of people (assigned to roles) with formal authority to make decisions (e.g., [2–4,26–29]) | Sequence of levels constructed by people in terms of perceived differences in e.g., seniority, age, experience or expertise (e.g., [11–14,23,37,42,43]) | Sequence of decision/task domains to which people have an intrinsic sense of obligation and commitment (e.g., [6–8,44–46]) | Sequence of levels in which people establish themselves as leaders by invoking an ideology to justify the hierarchical relationships between higher and lower levels (e.g., [5,52–57]) |
| *Examples* | Board of directors \| CEO \| unit managers \| heads of department \| etc. | Experienced employee \| junior employee (typically, within same unit/team) | Employees that start and/or join a bottom-up initiative to develop a new corporate strategy; members of a newly formed worker cooperative who nominate themselves and are then elected as managers of this cooperative | Leader–follower hierarchy emerging from a strong shared belief that the leader, for example: <br> - (in a political party) has the vision that can create political change and momentum; <br> - (in a Buddhist organization) has access to a higher level of spirituality; <br> - (in Roman-Catholic church) deputizes for God |
| *Core concept* | *Authority:* the legitimate power to make decisions | *Status:* One's relative social standing or professional position, that is, the respect one has in the eyes of others | *Responsibility:* The sense of intrinsic obligation to oneself, others and/or particular challenges | *Ideology:* The prevailing (e.g., religious, spiritual or political) values and beliefs regarding how the organization should operate |
| *Social mechanism* | Legitimacy of authority, as it arises from the constitution (or statutes) of the organization | Social stratification: Social construction of achieved status differences, drawing on shared cultural beliefs that make status differences appear natural and fair | Self-organization of responsibility, in which individuals take charge of tasks/challenges at higher levels of abstraction | Creating, adopting and/or sustaining a strong ideology, which operates as a cluster of (implicit) values and imperatives that "bracket" the ways in which members of the organization should think and operate |
| *Assumptions* | Decision-making authority is (initially) concentrated at the top, which may delegate authority to lower levels to reduce (consequences of) information overload and bounded rationality | Source of status is contingent on what drives respect and deference for other people within the (same unit of the) organization | Responsibility is something that people "take" rather than "get", in order to grow and sustain a substantial level of intrinsic obligation and commitment | Ideologies influence how people make sense of their (organizational) world, by providing standardized interpretations of the environment and thereby reducing uncertainty |

More generally speaking, a ladder of ideology is a sense-making mechanism that creates and sustains a set of collective beliefs and values, which in turn provide standardized interpretations of the environment and thus reduce uncertainty [59]. Any ideology is a black box involving a cluster of (mostly implicit) values and imperatives that serve to "bracket" the ways in which members of the incumbent organization should think and operate [60]. Compared to the other hierarchy types, the ladder of ideology is thus much more tacit and obscure.

*3.5. Overview*

Table 1 provides an overview of the four types of hierarchy identified in the literature. As such, this typology incorporates the well-known distinction between (formal) authority-based and (informal) status-based hierarchy [11,12,55] but also goes beyond this distinction by defining two other types.

The overview in Table 1 refers to each hierarchy type in its *archetypical* form. In practice, the hierarchy prevailing in many organizations tends to involve mixed instantiations of these archetypes. For example, several studies have demonstrated how "visionary" leaders (on top of the authority ladder) select other managers based on their fit with a core ideology as well as thoroughly indoctrinate employees into this ideology, to create a strong cognitive framework that drives employee behavior [57,61] or how top management's ideology affects the way subordinates make sense of key problems and opportunities [62,63]. In these organizations, the instantiated hierarchy, therefore, appears to involve both authority-based and ideology-driven ladders. Other examples are how worker cooperatives over time tend to extend and integrate their initial ladder of responsibility with ladders of authority and status [50,64].

## 4. Further Development of the Typology

This section serves to further develop the typology. First, the four hierarchy types will be mapped in terms of the tacit/explicitness of the knowledge constituting them. Subsequently, the four types are categorized using two additional dimensions.

Using knowledge theory [65], the four types of hierarchy can be placed on a continuum from fully tacit to fully explicit knowledge: see Figure 1. The ladder of authority is the most explicit form of hierarchy, with written rules and procedures as a defining characteristic [30,66,67]. These written rules originate from the constitution and statutes of the organization, extended via (executive) decisions on lower-level decision domains—communicated via job descriptions, decision logs, meeting minutes and other texts [3]. Such written rules on decision authorities impose normative and behavioral restrictions on subordinates [66]. In many instances, these rules are followed deliberately and consciously, while in other instances "rule following in organizations occurs unnoticed because rules have been internalized, have become unconscious premises of action or have been incorporated into firmly established and widely practiced routines and procedures" ([66], p. 9).

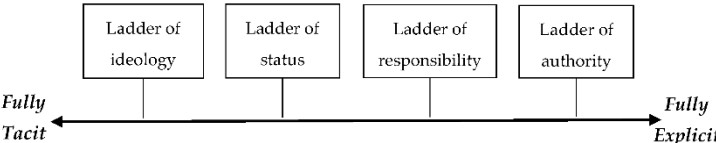

**Figure 1.** Four types of hierarchy on the tacit-explicit knowledge continuum.

At the other end of the continuum in Figure 1, ideology ladders appear to be largely tacit in nature because they draw on so-called collective tacit knowledge [65]. As observed in Section 3.4, a ladder of ideology is a sense-making mechanism that creates and sustains a set of collective beliefs and standardized interpretations, which operate (especially for outsiders) as a black box filled with tacit and obscure knowledge [59,60].

Status and responsibility ladders are positioned in the middle of the continuum in Figure 1. A status ladder draws a bit more on somatic knowledge, arising from the properties of individual bodies and brains as physical entities [65], whereas responsibility ladders appear to be somewhat more explicit in nature [8,50,51] than their status-based counterparts.

Another way to map the four types of hierarchy draws on the (in)transitive nature of each type as well as its behavior/cognition-centeredness; see Figure 2. The notion of transitivity refers to the extent to which the social mechanism (e.g., authority or status) can be delegated and/or transferred from one level to another [7]. In this respect, the authority and ideology ladder are both transitive in nature. That is, formal authority and strong

ideology can be relatively easily delegated or cascaded from the top level to various lower levels. As a result, large corporations, governmental systems and religious organizations tend to have rather deep hierarchies.

By contrast, responsibility and status cannot be (easily) delegated or transferred to other people, and these ladders are therefore non-transitive. Accordingly, responsibility and status ladders are unlikely to have more than two layers. For example, when persons A and B share the perception that A has a higher (e.g., expertise-driven) status, and B and C both believe B has a higher status, it does not follow that A and C also have a common perception of their relative status. A similar argumentation applies to a ladder of responsibility, given that an intrinsic commitment to a particular challenge cannot be (easily) transferred from one person to another: that is, to take charge of a higher-level challenge, individual employees have to climb up the ladder themselves because they cannot transfer their sense of responsibility to someone else [7]. Therefore, any ladder of responsibility or status is likely to have only two levels.

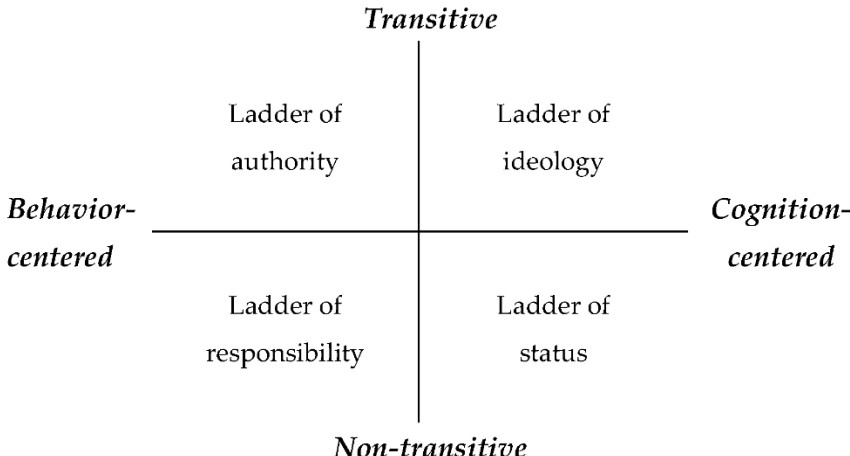

**Figure 2.** Types of hierarchy categorized in terms of their transitive/non-transitive nature and behavior/cognition-centeredness.

The other axis in Figure 2 is the difference between a behavior-centered and cognition-centered hierarchy. A behavior-centered ladder focuses on actions (to be) constrained or taken. By contrast, a cognition-centered hierarchy focuses on the mental activity required to carry this hierarchy, including various attentional, judgmental, reasoning, sensory and neural processes [68]. Here, the ladders of authority and responsibility are both largely centered around behavior: the authority-based ladder draws on decision domains, authorization procedures, budget constraints and actual decisions taken (e.g., [3,66]) and the ladder of responsibility involves people seeking higher-level responsibilities and acting accordingly (e.g., [7,69,70]). By contrast, ideology-based and status-based hierarchies are both largely cognition-centered, by either invoking some ideology to legitimize the hierarchical relationship between higher or lower levels (e.g., [5,52–55]) or drawing on shared cultural beliefs regarding status differences (e.g., [13,14,23,39]). The two dimensions together result in the matrix in Figure 2.

## 5. Discussion and Conclusions

In the two preceding sections, I developed a typology by reviewing and synthesizing the various ways in which hierarchy as a key characteristic of any complex system can be conceived. The ladders of authority, status, responsibility and ideology are archetypes of hierarchy that can be used as ideal templates [30] for coding and interpreting data. From the perspective of the formal-informal dichotomy [11,12,15], one might argue that the responsibility-based hierarchy can be subsumed as a special case of the formal ladder of authority, and the ideology-based hierarchy is a special case of the informal ladder of status.

However, the review in Section 3 and the resulting typology in Table 1 demonstrate that the core concepts and social mechanisms of these four types are sufficiently distinctive.

The typology developed in this paper has major implications for research on organizational and other social systems, which thus far tends to draw on a single conception of hierarchy (e.g., [14,42,71,72]) or focuses on the interaction between authority-driven and status-based hierarchy (e.g., [11,12,15]). Here, the broader mechanism-based framework outlined in Table 1 can guide future research efforts on various kinds of systems. More specifically, future work in the area of organizational citizenship [73,74], emergent leadership in self-managing teams [75,76], power and empowerment in social systems [77,78] and new organizational forms [79,80] can greatly benefit from a more differentiated understanding of the various ways in which hierarchy can arise. For example, scholars studying new organizational forms can develop theories of the interaction and integration of multiple types of hierarchies outlined in Table 1, also to resolve longstanding disputes on the nature and role of hierarchy in modern organizations [44,55,81–85]. In doing so, the three constructs used in Figures 1 and 2—knowledge explicitness, transitivity and behavior- versus cognition-centeredness—can be turned into scales for coding qualitative data, developing survey questions, and so forth.

Moreover, the distinctive characteristics of each type, outlined in Figures 1 and 2, can be highly useful in understanding why particular types of hierarchy prevail in specific contexts. For example, the notion of transitivity helps explain why ladders of authority and ideology are widely used (and combined) in large organizations, where decision-making authority and strong ideology can be effectively cascaded by means of deep ladders (e.g., [3,5]). Similarly, the intransitive nature of status and responsibility ladders explains why these ladders primarily exist at the micro-level of many organizations (e.g., [13,14]). In general, the four types of hierarchy appear to be highly complementary, in terms of their tacit/explicitness, (in)transitivity and behavior/cognition-centeredness—which helps more deeply understand their co-existence and possibly integration in practice.

Notably, the four types are best conceived as archetypes, that is, theoretically pure types that can be empirically combined and synthesized. Consequently, each quadrant in Figure 2 does not represent a silo but reflects a pure type of hierarchy that empirically often co-exists with several other types. The co-existence and co-evolution of multiple types of hierarchy in empirical settings provides a highly promising avenue for further research. Here, future work on specific social systems should seek to deeply understand the complementarity between different hierarchy types as well as the (potential) dominance of one type over others.

The multi-faceted nature of hierarchy also reflects the fact that some form of hierarchy exists in human as well as nonhuman systems [12,55,86–88]. That is, hierarchy appears to be functionally adaptive in allowing any kind of group "to achieve the high levels of coordination and cooperation necessary to ensure survival and success" ([87], p. 33). In historical terms, hierarchy has deep roots in our social systems, from the relatively simple ladders prevailing in hunter-gatherer groups (about 2 million years ago) to the very elaborate and complex hierarchies we are witnessing today. Hierarchies in pre-historic and ancient times were largely based on status and ideology [53,54,86], whereas social systems today appear to increasingly thrive on ladders of authority and responsibility. Accordingly, human civilization appears to gradually evolve from more tacit forms of hierarchy to more explicit ones (cf., Figure 1) as well as from cognition-centered to more behavior-centered types of hierarchy (cf., Figure 2). This long-term evolution illustrates the plasticity of the hierarchy construct in genealogical terms.

The typology developed in this article underlines the functional adaptability of hierarchy but also serves to clarify the pivotal role of the hierarchy metaphor in systems theory (e.g., [89,90]). In this respect, the broad notion of hierarchy envisioned by Simon [10,85] appears to be especially problematic when its underlying mechanisms are not properly defined and understood. The typology developed in this article dissects the general metaphor of hierarchy into four types, each with a distinct social mechanism.

**Funding:** This research received no external funding.

**Institutional Review Board Statement:** Not applicable.

**Informed Consent Statement:** Not applicable.

**Conflicts of Interest:** The author declares no conflict of interest.

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
