# Peer review of "Ladders of Authority, Status, Responsibility and Ideology: Toward a Typology of Hierarchy in Social Systems"

_systems, doi:10.3390/systems9010020_

Round 1

Reviewer 1 Report

In my opinion this is a great conceptual paper. I think it analyzes in deep the concept of hierarchy in complex systems.

I liked very much the review made by the author and overall the typology proposed.

I think is a good reflexion and viewpoint that can be used as a base for future analysis (theoretical or empirical).

Author Response

Many thanks for your positive assessment of this manuscript.

Reviewer 2 Report

The author has organized voluminous and fragmented literature on the subject of hierarchy into an important typology relevant to research on the design and performance of organizations. The paper is well written and easy to follow. I highly recommend it for publication. I have following suggestions for a minor revision:

  1. The author mentions in the second section that the focus of the typology is organizations, which should reflect in the title. I suggest changing the second phrase in the title to: Towards a Typology of Hierarchy in Organizations.
  2. Figure 1 proposes an interesting scale that can potentially be tied to organizational performance or the type of service it might deliver. A reference to the use of this scale in suggestions for further research would be desirable.
  3. Figure 2 points to another interesting space in which an organization might be located with respect to its performance, function and leadership style. I surmise the quadrants are not like silos but reflect dominance of a specific ladder along with the presence of others, which I recommend clarifying. I also recommend reference to this space in suggestions for further research.

Author Response

Many thanks for positively assessing this manuscript and recommending it for publication. I have addressed your comments as follows:

  1. You suggest to change the second phrase in the title to: "Towards a Typology of Hierarchy in Organizations." Notably, while the literature review in this paper indeed largely focuses on organizational hierarchy, I did include a few examples of other social systems. So, I kept the title as it is.  To clarify this, I did add a sentence to the second paragraph of section 2 (lines 75-77), which says: "This also implies that the next section contains several examples of (hierarchy used in) social systems other than organizations."
  2. This is a very good suggestion. In fact, your comment also applies to the two dimensions used in Figure 2. Accordingly, I inserted the following sentence in the second paragraph of the Discussion & Conclusion section (lines 320-323): "In doing so, the three constructs used in Figures 1 and 2—knowledge explicitness, transitivity, and behavior versus cognition centeredness—can be turned into scales for coding qualitative data, developing survey questions, and so forth."
  3. In response to your third suggestion, I extended and rewrote the last sentence of the third paragraph in section 5 into a new (fourth) paragraph (336-342): "Consequently, each quadrant in Figure 2 does not represent a silo but reflects a pure type of hierarchy that empirically often co-exists with several other types. The co-existence and co-evolution of multiple types of hierarchy in empirical settings provides a highly promising avenue for further research. Here, future work on specific social systems should seek to deeply understand the complementarity between different hierarchy types as well as the (potential) dominance of one type over others."

Reviewer 3 Report

First of all, thank you for the opportunity to review this suggesting work, which in my opinion clearly deserves to be published.

This article contributes to rethinking, from an academic perspective, the hierarchy concept applied to the organizational systems. From a qualitative and interpretive approach, papers that have been published under a double bind reviewed process are reviewed in a systematic and iterative way. The result of the study allows to establish 4 types of hierarchy, and what is more relevant, the implicit assumptions in each one of the types for the study of organizations. Likewise, I think that the explicit recognition that archetypes are involved, which can be combined, is a very relevant precision.

In short, I think it is a very original and relevant article and, as I have said, it deserves to be published and taken into account in organizational studies. Personally, the revision has provided me with insights for reflection on my research practice in the field of high-risk organizations.

Nothing to object, just suggestions that might help to visualize the strength of the results, in other word, the theoretical-practical relevance of the results. They are issues related to making the analysis process more visible.

- It would be interesting to have a table systematizing the findings of the published articles, with some type of information (type of journal, subject, year ... number of articles)

- Brief information indicating how the articles have been qualitatively reviewed.

- Would it be possible to establish a temporal evolution of the types of hierarchy in the studies? Perhaps some reflection would be interesting in the discussion about the plasticity of the concept in genealogical terms.

Author Response

Many thanks for your highly constructive review and assessment. I have revised the manuscript in response to your comments as follows.

  • You suggest inserting a table that outlines the findings of the published articles, with some type of information (type of journal, subject, number of articles, etc.). I have considered doing this, but decided not to add an additional table. The key findings from the review are properly outlined in Table 1 (including a list of key sources in the first row of this table). In response to your suggestion, I might add the observation that the authority and status ladders are predominantly used/reported in organization, management, law and economics journals, whereas the two other hierarchies have been primarily identified in journals and books on communication science, sociology, anthropology, and the like. (Which will also become clear for readers, if they go to the sources listed at the end of the article.) Overall, I feel this does not add any clarity to the argument and typology in this paper.
  • You suggest to provide concise information about how publications have been qualitatively reviewed. Accordingly, I added several sentences about the review approach to the second paragraph of section 2, for example (lines 83-84): "Each publication was reviewed and coded regarding the definition, social mechanisms, and assumptions of hierarchy." 
  • Finally, you suggest to identify a temporal evolution of the types of hierarchy. Accordingly, I've inserted the following new text in the final section of the article (lines 346-355): "In historical terms, hierarchy has deep roots in our social systems, from the relatively simple ladders prevailing in hunter-gatherer groups (about 2 million years ago) to the very elaborate and complex hierarchies we are witnessing today. Hierarchies in pre-historic and ancient times were largely based on status and ideology [53,54,86], whereas social systems today appear to increasingly thrive on ladders of authority and responsibility. Accordingly, human civilization appears to gradually evolve from more tacit forms of hierarchy to more explicit ones (cf., Figure 1) as well as from cognition-centered to more behavior-centered types of hierarchy (cf., Figure 2). This long-term evolution illustrates the plasticity of the hierarchy construct in genealogical terms."